# Bilateral Thygeson’s Superficial Punctate Keratitis with Dendritic Corneal Lesion: A Case Report

**DOI:** 10.3390/medicina59010163

**Published:** 2023-01-13

**Authors:** Woo Seok Choe, Tae Gi Kim

**Affiliations:** Department of Ophthalmology, Kyung Hee University College of Medicine, Kyung Hee University Hospital at Gangdong, Seoul 05278, Republic of Korea

**Keywords:** herpes keratitis, herpes simplex, herpes zoster, Thygeson’s superficial punctate keratitis, cyclosporine A

## Abstract

Thygeson’s superficial punctate keratitis (TSPK) is a recurrent bilateral corneal epithelial disease. Typically, small, multiple discrete epithelial lesions occur in the central cornea. However, dendritic corneal lesions are rare. Herein, we report a rare case of TSPK in both eyes after a unilateral dendritic corneal lesion. A 42-year-old woman presented with decreased vision and foreign body sensation in her right eye that persisted for 1 month. Her uncorrected visual acuity and best-corrected visual acuity (BCVA) were 20/160 in the right eye. Slit-lamp microscopy revealed a dendritic lesion in the central cornea of the right eye. No abnormalities were observed in her left eye. Herpetic keratitis in the right eye was diagnosed and systemic acyclovir was prescribed, along with topical acyclovir ointment and steroids. After one week, most of the corneal lesions had disappeared, and the BCVA in the right eye had improved to 20/25. The corneal epithelium completely recovered after 2 weeks. However, 2 weeks later, the patient visited the hospital with decreased visual acuity in the right eye, and the BCVA decreased to 20/40. Multiple fine corneal lesions were observed under a slit-lamp microscope. The patient was diagnosed with TSPK of the right eye. Topical steroids were started, and after 7 days, the corneal condition improved. However, after 6 weeks, visual acuity decreased in the left eye, and a corneal lesion similar to that in the right eye was observed; therefore, the patient was diagnosed with bilateral TSPK. Short-term topical steroids and long-term topical cyclosporine A 0.1% were used in both eyes, and the disease was maintained without recurrence for 3 months. TSPK can appear as a unilateral dendritic corneal lesion similar to herpetic keratitis. Therefore, in case of unilateral dendritic corneal lesions, it should be considered that TSPK may develop later.

## 1. Introduction

Thygeson’s superficial punctate keratitis (TSPK) is a recurrent bilateral corneal epithelial keratopathy first described in 1950 [1]. Patients with TSPK may experience repeated corneal irritations and blurred vision. TSPK primarily affects the central cornea and typically shows focal and discrete round-to-oval elevated epithelial lesions that show negative staining with fluorescein on slit-lamp examination.

Although the exact etiology of TSPK remains unclear, an association between viral and immunogenic components is hypothesized. In early TSPK studies, many researchers agreed that the cause of TSPK was a virus; however, they were unable to directly identify the virus through electron microscopy and polymerase chain reaction (PCR) [2,3,4]. Reinhard et al. obtained corneal epithelial cells from nine TSPK patients and analyzed the varicella zoster virus (VZV) genome, but failed to amplify the DNA of the VZV genome [2]. Connell et al. attempted to detect other viruses, such as herpes simplex virus (HSV) types 1 and 2, VZV, and adenovirus, but concluded that these viruses did not cause TSPK [5]. In contrast, Darrell and Suciu-Foca reported that TSPK was related to immune response, focusing on the high frequency of detection of the HLA-DR3 antigen in patients with TSPK [6]. However, Quere et al. concluded that the disease is associated with an allergic reaction based on the fact that TSPK responds well to steroids [7].

Although the cause of TSPK is suspected as virus-induced antigen-antibody complexes or T-cell-mediated immune responses, few reports of cases of TSPK occurring after dendritic corneal lesions suspected of viral infection have been documented. Most previous cases of TSPK have been reported to have multiple punctate corneal lesions; however, cases of dendritic corneal lesions are very rare [1,3,4,6]. Here, we report a rare case of typical bilateral TSPK occurring after the development of a unilateral dendritic corneal lesion similar to herpes keratitis.

## 2. Case Report

A 42-year-old woman presented to our clinic with complaints of decreased vision and foreign-body sensation in her right eye. The patient was diagnosed with herpetic keratitis in the right eye at a local clinic 1 month prior and was treated with acyclovir ointment five times a day. The patient had no history of systemic immunosuppression.

On ophthalmic examination, her uncorrected visual acuity (UCVA) and best-corrected visual acuity (BCVA) were 20/160 and 20/160, respectively, in the right eye and 20/100 and 20/20, respectively, in the left eye. The intraocular pressure (IOP), measured using a non-contact tonometer, was 23 mmHg in the right eye and 24 mmHg in the left eye. Slit-lamp examination revealed a dendritic corneal lesion similar to a dendritic ulcer in the central cornea of the right eye, with no abnormality in the left eye (Figure 1A). The patient was diagnosed with keratitis caused by either herpes zoster or herpes simplex virus. She was prescribed topical acyclovir ointment five times a day, topical fluorometholone 0.1% four times a day, and 400 mg systemic acyclovir four times a day for 7 days.

One week after the initial presentation, the BCVA of the right eye improved to 20/25. Corneal pseudodendritic lesions almost disappeared under slit-lamp examination, and only mild punctate corneal erosion was observed. Systemic acyclovir was discontinued, and topical acyclovir ointment and fluorometholone 0.1% were switched to twice daily use. After 2 weeks, the BCVA in the right eye improved to 20/20, and the corneal epithelium completely recovered (Figure 1B). All medications were discontinued, and the patient was scheduled to visit the hospital again after a month.

However, 2 weeks later, the patient revisited the hospital because of decreased visual acuity in the right eye. The BCVA of the right eye had decreased to 20/40, and central corneal lesions with multiple elevated lesions with negative staining were observed under a slit lamp examination (Figure 1C); the left eye showed no abnormality and its BCVA was 20/20. TSPK was confirmed in the right eye, and use of topical fluorometholone 0.1% was started four times a day. Topical cyclosporin A 0.1% was added once daily. After 7 days, the cornea and visual acuity improved to 20/20 (Figure 1D). Use of topical fluorometholone 0.1% was discontinued and topical cyclosporine A 0.1% was continued in the right eye once a day for 3 months.

After 6 weeks, the patient presented with decreased visual acuity in the left eye, along with a decrease in the BCVA to 20/20. A central corneal lesion with multiple elevated lesions similar to those in the right eye was also observed in the left eye (Figure 2A). Finally, bilateral TSPK was confirmed, and topical fluorometholone 0.1% four times a day in the left eye and topical cyclosporine A 0.1% in both eyes were used for 1 week. After 1 week, the lesion in the left eye also improved, without recurrence in the right eye; the BCVA improved to 20/20 in the left eye and was maintained at 20/20 in the right eye (Figure 2B). No recurrence was observed after 3 months of treatment with 0.1% cyclosporin A (Figure 2C,D).

## 3. Discussion

TSPK is a chronic, recurrent, bilateral corneal disease caused by focal epithelial keratitis without significant conjunctival or stromal inflammation [3,6]. Typically, multiple punctate and whitish lesions are observed in the corneal epithelium. However, in our case, a dendritic lesion was observed, which is very rare *in* TSPK. Darrell et al. previously reported a case of TSPK with a branched or pseudodendritic appearance of atypical corneal lesions, but dendritic lesions were not clearly visible [6].

The pathophysiology of TSPK remains unclear, but it is thought that a viral or immune response induced by a viral infection may trigger TSPK because of the prolonged episodes of exacerbations and remissions, and monocyte reaction. In our case, TSPK occurred bilaterally after unilateral dendritic corneal lesion, which suggests the possible involvement of viral and immunogenic components in the development of TSPK. The timeline of our case is presented in Figure 3.

It is suspected that TSPK has a viral etiology; however, this has not been proven for TSPK [4]. Several studies have reported cases of viral infection in patients with TSPK. Braley and Alexandra isolated a single strain of virus from the corneal epithelium of a rabbit but failed to detect the virus in subsequent attempts [8]. In addition, Lemp et al. isolated varicella zoster virus from the corneal epithelium of TSPK patients; however, the results have not been reproduced [9]. On the other hand, in several studies using tissue culture techniques, a viral infection was not detected in TSPK patients. Connell et al. attempted to detect epithelial viral infection in TSPK patients through polymerase chain reaction and reported the absence of HSV type 1,2, HZV, and adenovirus [5]. In our case, a unilateral dendritic lesion occurred before typical clinical features of bilateral TSPK appeared. TSPK should be carefully differentiated from other types of keratitis, especially HSV keratitis. TSPK is often misdiagnosed as viral keratitis because the lesions associated with TSPK are similar to the corneal infiltrates associated with viral keratitis. TSPK presents with multiple central corneal lesions without conjunctival hyperemia or discharge, whereas HSV patients typically have a painful red eye, with either a diffuse mild epithelial haze or dendritic ulceration. Although dendritic lesions are rarely seen in the early stage of TSPK, as in our case, it is easy to misdiagnose as viral keratitis. In addition, TSPK may appear similar to the subepithelial infiltration observed in adenovirus keratitis.

Another hypothesis is that TSPK occurs because the histocompatibility antigen (HLA) DR3 alters the immune response to exogenous or endogenous viral infections in TSPK patients [6,10]. Lymphocyte response to steroids in the corneal epithelium, prolonged course, exacerbations and remissions characteristic of the disease, and T-cell-related immune mechanisms play an important role in TSPK, which supports the theory of an immunological component underlying this condition. Quere et al. reported that an allergic reaction may trigger TSPK and suggested that some TSPK patients had systemic allergic disease and responded well to steroids [7]. However, this proposition is disputed because eosinophils are not detected.

Topical corticosteroids are the most widely used treatment for TSPK, and in our case, symptom improvement was achieved with the initial steroid use. However, recurrence is common once steroids are withdrawn, and there are reports that a prolonged course of disease may be related to the use of corticosteroids [2,11]. Recently, it was reported that immunomodulatory agents can be used effectively in TSPK instead of corticosteroids [10]. In our case, the use of topical cyclosporine A, an immunosuppressive agent, resulted in long-term improvement of the lesion without recurrence, indicating that stabilizing the immune response through the use of long-term immunosuppressive agents is important for TSPK treatment.

This case report presents a limitation in that additional tests were not performed to isolate the virus. For TSPK, there is no specific diagnostic test, and the diagnosis is based on clinical features, slit-lamp examination findings, and therapeutic response to topical corticosteroids. In addition, a PCR test may only prove an absence of viral infection as was seen in several previous studies. Therefore, it is difficult to conclude whether the initial lesions seen in our case were caused by HSV or simply resembles HSV keratitis. However, our case showed a unilateral dendritic corneal lesion at early stage, which is rare in TSPK and very similar to the viral disease. We thought that our report is meaningful as a case in which a viral origin should not be completely excluded as one of the causes of TSPK.

## 4. Conclusions

In conclusion, we report a rare case of TSPK that occurs bilaterally after a unilateral dendritic corneal lesion. The case is meaningful because it corroborates the hypothesis that viral etiology may underlie the pathogenesis of TSPK. Because dendritic corneal lesions may be seen in the early stages of TSPK, it is important to consider the possibility of TSPK while distinguishing it from other types of viral keratitis.

## Figures and Tables

**Figure 1 medicina-59-00163-f001:**
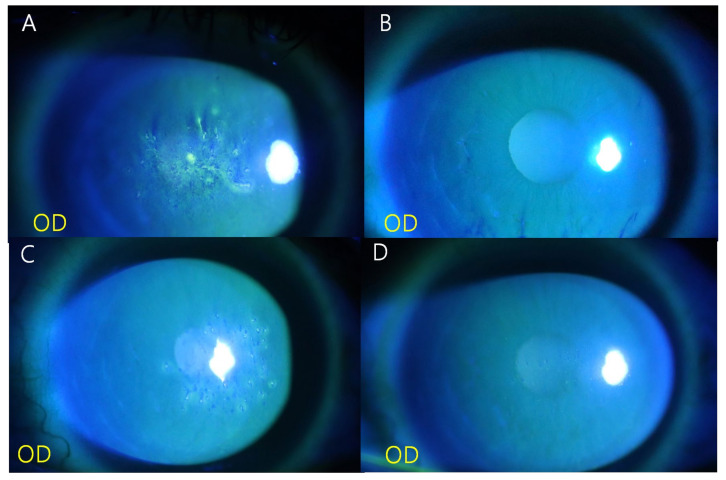
At the initial visit, a dendritic corneal lesion was observed in the right eye (**A**). Three weeks after the initial visit, complete recovery of the corneal epithelium is noted (**B**). At 5 weeks, elevated cornea lesions with a discrete round to oval shape scattered diffusely are noted in the patient’s right eye (**C**) and complete recovery of the corneal epithelium is noted at 10 weeks (**D**).

**Figure 2 medicina-59-00163-f002:**
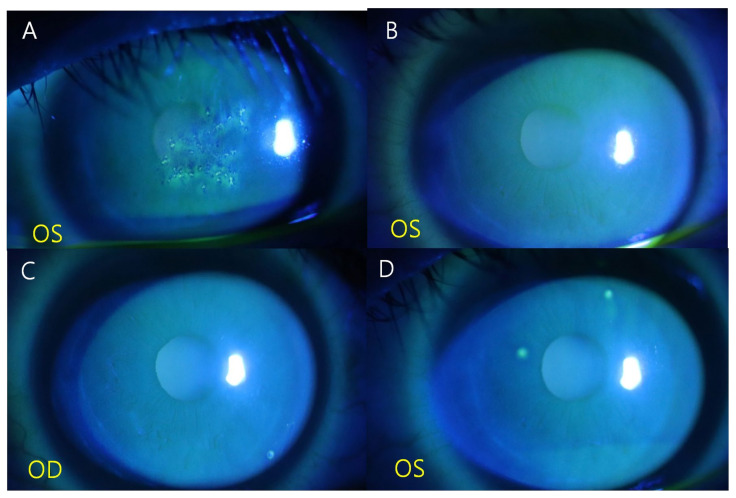
At 16 weeks after the initial presentation, the central corneal lesion with multiple elevated lesions was observed in the left eye (**A**) and the corneal lesion improved after 1 week (**B**). No recurrence was observed 3 months after the last event in both eyes (**C**,**D**).

**Figure 3 medicina-59-00163-f003:**
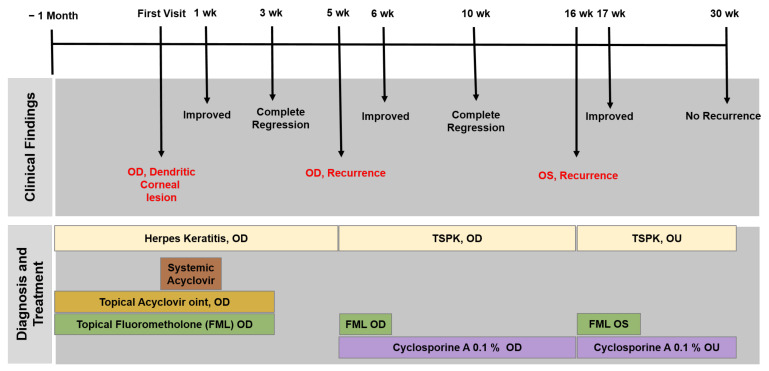
Timeline of the reported case.

## Data Availability

Not applicable.

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
