# Peer review of "Bilateral Thygeson’s Superficial Punctate Keratitis with Dendritic Corneal Lesion: A Case Report"

_medicina, 2023, doi:10.3390/medicina59010163_

Round 1

Reviewer 1 Report

The manuscript is well written on an interesting case. The reviewer has no further comments or suggestions for the authors.
1. This is a short case report on two rare eye diseases occurring simultenously in the same patient.
2. For eye doctors its is important to know how to organize the follow-up of these patients at clinics.
3. Although the keratitis thygesonii was reported for the first time more than 70 years ago, we don't know the etiology of this eye disease and how it relates to other eye diseases. A good summary on what has been suggested to be behind this eye disease.
4. The figures describe clearly the findings.
5. The conclusions are appropriate and consistent with the evidence presented.
6. To my mind, the references are appropriate. 7. No additional comments.

Author Response

The manuscript is well written on an interesting case. The reviewer has no further comments or suggestions for the authors.

1. This is a short case report on two rare eye diseases occurring simultenously in the same patient.2. For eye doctors its is important to know how to organize the follow-up of these patients at clinics.3. Although the keratitis thygesonii was reported for the first time more than 70 years ago, we don't know the etiology of this eye disease and how it relates to other eye diseases. A good summary on what has been suggested to be behind this eye disease.4. The figures describe clearly the findings.5. The conclusions are appropriate and consistent with the evidence presented.6. To my mind, the references are appropriate. 7. No additional comments.

Response: We are grateful to you for these insightful comments. We also thank you for these encouraging words.

Reviewer 2 Report

Summary

In this article the authors describe in detail the rare case of a patient with bilateral superficial persistent Thygeson's punctate keratitis diagnosed after suspected unilateral HSV keratitis.  According to the authors' description, the clinical picture of the latter keratitis (dendritic lesion typical of HSV involvement) which appeared to be Thygeson's keratitis provides, at least, arguments on a similarity of mechanisms between the two pathologies or even links between them. This study may be of great interest for the understanding of Tygeson's superficial punctate keratitis.

General concept Comments:

The study is very well described with a background with many references and essential information.
The paper is clear, explaining visit after visit, the therapeutic path of this patient during this episode of Thygeson's punctate keratitis.
The results show that this condition seems to be related to HSV keratitis with a discussion highlighting the different hypotheses and the (very important) limitation of this study.
It is unfortunate that the patient did not have a PCR test at his first visit to rule out HSV infection, especially as treatment seems to have been effective after this diagnosis.
The conclusion reached is well supported by the results and it is important to place HSV as one of the hypotheses but not the only valid hypothesis!

The questions that can be asked following the argument are the following:

- Has the patient had HSV keratitis or any other symptom of HSV infection in the past?

- Is the first diagnosis of HSV keratitis really wrong? Without PCR, it is difficult to be sure. What do you think? Also the patient did have regression of the keratitis following treatment of it (topical + systemic Acyclovir, FML) which is different to the treatment of TSPK (Cyclosporine, FML). Is this related to chance?

- Does TSPK have an etiological link to HSV infection? OR is TSPK a condition with similar mechanisms to HSV infection?"

- Are one pathology resembling another necessarily related? Could it be that only the mechanism of action is similar and not the causal link? Beware of confirmation bias!

Author Response

The study is very well described with a background with many references and essential information. The paper is clear, explaining visit after visit, the therapeutic path of this patient during this episode of Thygeson's punctate keratitis. The results show that this condition seems to be related to HSV keratitis with a discussion highlighting the different hypotheses and the (very important) limitation of this study. It is unfortunate that the patient did not have a PCR test at his first visit to rule out HSV infection, especially as treatment seems to have been effective after this diagnosis. The conclusion reached is well supported by the results and it is important to place HSV as one of the hypotheses but not the only valid hypothesis! 

Response: We are grateful to you for these insightful comments. We also thank you for these encouraging words.

The questions that can be asked following the argument are the following:

Point 1: Has the patient had HSV keratitis or any other symptom of HSV infection in the past?

Response 1: The patient had no history of symptoms related to HSV keratitis. The patient's symptoms started first one month before visiting our clinic, and she treated with topical acyclovir at the local clinic.

Point 2: Is the first diagnosis of HSV keratitis really wrong? Without PCR, it is difficult to be sure. What do you think? Also the patient did have regression of the keratitis following treatment of it (topical + systemic Acyclovir, FML) which is different to the treatment of TSPK (Cyclosporine, FML). Is this related to chance?

Response 2: We thank you for this suggestion. It is difficult to determine if a patient actually had HSV keratitis. According to the patient's past history, she was diagnosed with HSV keratitis at a local clinic, and topical acyclovir ointment was used for one month, but there was no improvement. However, it is difficult to conclude that HSV is not the cause of corneal lesions even if there is no response to anti-viral treatment. Although HSV keratitis is initially related to viral proliferation, the immune response eventually aggravates corneal lesions, and steroid use can be helpful. For this reason, topical steroids were also used in our clinic. Since TSPK is also a disease caused by an immune response, the initial lesion regression is thought to be the effect of steroids. However, after discontinuation of eye drops, multiple corneal lesions recurred in a short time, there were no accompanying symptoms such as conjunctival hyperemia, and it also occurred in the left eye without lesions, so TSPK seems appropriate for the patient's final diagnosis. 

Point 3: Does TSPK have an etiological link to HSV infection? OR is TSPK a condition with similar mechanisms to HSV infection?"

Response 3: Initially, TSPK was thought to be a disease caused by HSV, HZV or adenovirus, and studies were conducted on this. However, they failed to isolate the virus, and the cause of TSPK is still unclear. Reasons for presuming that the cause of TSPK is a virus include the absence of bacteria, the resistance to antibacterial agents, and mononuclear cell exudate on conjunctival scrapings. Also, TSPK has several features in common with known viral infections of the cornea such as intraepithelial lesions, the long duration, exacerbations and remisions, and a mononuclear cell response. However, TSPK did show cell destruction confined to one discrete area, which contrasted with the cell-to-cell spread of herpes simplex keratitis. As such, TSPK is considered a disease that has both characteristics that are shared with viral diseases and characteristics that are not.

Point 4: Are one pathology resembling another necessarily related? Could it be that only the mechanism of action is similar and not the causal link? Beware of confirmation bias!

Response 4: We agree with what you pointed out. As the you pointed out, there may be bias. Since the patient recurred in both eyes with multiple discreted corneal lesions after the primary treatment, the patient's diagnosis seems to be TSPK. However, it is not clear whether TSPK is triggered by HSV or the initial lesion simply resembles HSV keratitis. Nevertheless, the meaning of our case is that the symptoms started with an unilateral dendritic corneal lesion, which is rare in TSPK. And although it is difficult to conclude with only our case, the possibility of a virus as one of the causes of TSPK should not be completely ruled out through these clinical features. As you pointed out, we added this finding to the discussion section as follows.

Discussion (Page 5 Line 166 ):

"Therefore, it is difficult to conclude whether the initial lesions seen in our case were caused by HSV or simply resembles HSV keratitis. However our case showed unilateral dendritic corneal lesion at early stage, which is rare in TSPK and very similar to the viral disease. We thought that our report is meaningful as a case in which a viral origin should not be completely excluded as one of the causes of TSPK."